# Multivariate analysis for agro-morphological and quality traits in groundnut (*Arachis hypogaea* L.) genotypes in Eastern Ethiopia

**Desu Beriso Dama** ⬛\*, **Seltene Abadi Tesfamariam, Abdi Mohammed Hassen**

Department of Plant Sciences, Haramaya University, Haramaya, Ethiopia

\* desuberiso500@gmail.com

## Abstract

Groundnut is an important oil seed crop in Ethiopia, providing food, oil, and industrial products while contributing to soil fertility through nitrogen fixation. However, its production is limited by narrow genetic variability, low-yielding varieties, and environmental stresses, making it essential to assess genetic diversity among existing genotypes for effective breeding and improvement. The present study was carried out to assess the extent of genetic variability among groundnut genotypes for agro-morphological and quality traits. Thirty-six groundnut genotypes were evaluated in a 6 x 6 simple lattice design during 2023 post-rainy season under irrigation at Dire Dawa, the research station of Haramaya university, Ethiopia. Data were collected on kernel yield and other morphological traits, oil content and oil yield. The data on traits were subjected to *principal component* (*PC*) analysis, clustering and Euclidean distance. In this study, the first six *Principal Components Analysis* (*PCA*) found to be significant and accounted for 74.51% of the total variation in which the first *principal component* (*PC1*) and the second *principal component* (*PC2*) contributed more to the variation. PC1 and PC2, capturing most of the variation, identify the key traits contributing to genetic diversity, guiding the selection of distinct parents for effective groundnut breeding. Clustering of the genotypes resulted in six major clusters, and the dendrogram showed that cluster I, II, III, IV, V and VI included 6, 9, 8, 5, 7 and 1 genotypes, respectively. The result implies that, genotypes within the same cluster are genetically similar, while those in different clusters are more diverse, providing opportunities to select distinct parents for effective breeding. Euclidean distance ranged from 2.45 to 8.54 with the mean, standard deviation and coefficient of variation of 5.44, 1.17 and 21.56%, respectively. Based on the result of the current study, there were variations of genetic distances among genotypes, Gv17 and Gv28, Gv3 and Gv23, Gv3 and Gv30, Gv15 and Gv17, Gv22 and Gv28, and Gv3 and Gv34 which could be exploited through hybridization for cultivar development in groundnut breeding programs in Ethiopia. Therefore, the genetically divergent genotypes identified in this study provide valuable parental material for hybridization, offering strong

**Data availability statement:** All relevant data are within the manuscript and its Supporting information files.

**Funding:** The author(s) received no specific funding for this work.

**Competing interests:** The authors have declared that no competing interests exist.

potential for the development of improved groundnut varieties adapted to Ethiopian agro-ecologies.

## 1. Introduction

Groundnut, or peanut, is one of the significant oil seed crops, which is cultivated in tropical and subtropical regions of the world [1]. It belongs to the legume family *Leguminaceae* and is an annual plant originally native to South America, Mexico, and Central America, but is now successfully grown in a wide variety of environments on all six continents around the globe [2].

Globally, groundnut ranks as the fifth most produced oil seed in the world, after oil palm, soybean, rapeseed, and sunflower, and it is grown in more than 110 countries [3]. Asia accounts for the largest share of global production (49.84%) followed by the USA (32.35%) and Africa (13.78%). Major producing countries include China, India, Nigeria, the USA, and Sudan. Worldwide annual production is estimated at about 53.6 million tons from 31.6 million hectares, with an average yield of 1.70 t ha⁻¹. In Africa, groundnut is produced on about 17.43 million hectares with an average yield of 0.97 t ha⁻¹, while Eastern Africa contributes approximately 20.6 million tons from 2.8 million hectares, yielding 0.74 t ha⁻¹ [3].

Groundnut kernels are nutritionally rich, containing 35–56% oil, 25–30% protein, 9.5–19% carbohydrates, along with essential minerals and vitamins B, E, and K [4,5]. The oil quality is largely determined by fatty acid composition, particularly oleic and lineolic acids. High oleic acid content enhances shelf life, oxidative stability, and cardio vascular health benefits making oleic to lineolic acids ratio an important industrial quality indicator [6,7].

Groundnuts support diverse food and non-foods industries, including confectioneries, bakery products, paints, biofuels, lubricants, and insecticides [8]. In Ethiopia, groundnut is widely used in products such as groundnut butter, traditional sweets (*Haalawa*), beverages consumed during fasting, roasted snacks (*kolos*), and complementary foods such as *fafa* [9]. Additionally, groundnut-based therapeutic foods like Plumpy'Nut play a crucial role in combating child malnutrition [10]. Agronomically, groundnut contributes to soil fertility through biological nitrogen fixation and provides valuable by-products for livestock feed and biofuel [11,12].

The groundnut is considered the second most important lowland oilseed crop in Ethiopia after sesame, which has proven to be highly potential for oil crop expansion in the lowland regions of the country [4]. According to [13], about 205,068.65 tons of groundnuts are produced annually by the country, grown on 113,514.95 hectares and providing an average productivity of 1.807 tons per hectare. Oromia region dominates production, contributing over 59%, followed by Benishangul-Gumuz (24.83%), Amhara (7.43%), Harari (3.29%), and Southern Nations, Nationalities, and Peoples' Region (SNNP) contributing (1.29%). Eastern Hararghe Zone is the leading groundnut-producing and supplying area.

Despite its importance, groundnut production in Ethiopia is constrained by several biotic, abiotic, and socio-economic factors. Major challenges include drought stress,

particularly during flowering, soil-borne fungal diseases, poor post-harvest handling leading to aflatoxin contamination, low soil fertility, limited access to improved varieties, foliar diseases, and weak extension and credit services [14]. These constraints result in low productivity and reduced market competitiveness.

Farmers in Ethiopia prefer groundnut varieties with high pod and kernel yield, early maturity, drought tolerance, high market value, good seed quality, adaptability to local conditions, and resistance to root diseases and invasive weeds [15]. Addressing these demands requires the development of high-yielding, and resilient varieties with preferred traits.

Genetic improvement depends largely on the availability and effective utilization of genetic variability. Parameters such as heritability, genetic advance, and coefficient of variations are essential for identifying superior genotypes and selecting effective breeding strategies [3,5,1,16].

Cultivated groundnut possesses a narrow genetic base due to its monophyletic origin, self-pollinating nature, and limited gene flow with wild relatives [17]. The crop originated from a single hybridization event between two diploid species followed by chromosome doubling, restricting genetic recombination and variability [17]. This narrow genetic base poses challenges for yield improvement and stress tolerance.

Although groundnut plays vital role in eastern Ethiopia, comprehensive information on genetic divergence among Ethiopian groundnut genotypes particularly based on agro-morphological and quality traits remains limited.

Hence, there is an urgent need for a study of genetic variability among groundnut varieties for effective management, conservation, and optimal utilization of genetic resources in breeding programs. Genotype variation above groundnut varieties is critical in effective breeding processes, making it essential to evaluate existing variation in groundnut genotypes grown in Ethiopia. Despite the importance of groundnut in eastern Ethiopia, comprehensive information on genetic divergence based on agro-morphological and quality traits remains limited. Therefore, this study aimed to assess genetic diversity among groundnut genotypes using multivariate statistical approaches to identify divergent and superior parental lines for future breeding programs.

## 2. Materials and methods

### 2.1. Description of study area

The current study was carried out at Dire Dawa, Tony farm Research sub-Station of Haramaya University during 2023 post-rainy season under irrigation. The location is situated at 9° 35' 0" North and 41° 52' 0" East at an altitude of 1197 meters above sea level (m.a.s.l) in the eastern Rift Valley escarpment of Ethiopia. The average maximum and minimum temperatures of the area are 31.89 °C and 17.08 °C, respectively and mean annual temperature is 25.36 °C. The area receives average rainfall of about 637 mm per annum. Soils of the farm are predominantly sandy clay loam with pH varies with soil depth which is about 7.97 at a depth of 0–20 cm and about 8.44 at a depth of 20–40 cm. The bulk density range varies from 1.28 to 1.39 g cm $^{-3}$. The area is categorized as hot to warm arid according to agro-ecological belts of Ethiopia.

### 2.2. Experimental materials

Thirty-six groundnut genotypes, including two standard checks were obtained from the National Groundnut Program based at Haramaya University (Table 1).

### 2.3. Experimental design and field management

The field experiment was conducted using irrigation during the 2023 post-rainy season. The experiment was laid out in simple lattice design. Seeds of each genotype were sown in 3 rows of 3 meter long with spaces of 0.6 m between rows, 0.1 m between plants, and 0.5 m between plots. Hence, a total of 7.2 m² area was allocated for each experimental plot in each incomplete block within replication. Spacing between sub-blocks (incomplete block) and replications was 0.7 and 1 m, respectively. All agronomic practices were applied as per the standard procedure for groundnut production [18].

**Table 1. List of groundnut genotypes used in this study.**

| Codes for Genotype | Genotypes | Breeding status | Seed source |
|---|---|---|---|
| Gv1 | Babile-1 | Released variety | Haramaya University |
| Gv2 | Babile-2 | Released variety | Haramaya University |
| Gv3 | ICGV 00350 | Advanced line | ICRISAT-India |
| Gv4 | ICGV 02271 | Advanced line | ICRISAT-India |
| Gv5 | ICGV 03179 | Advanced line | ICRISAT-India |
| Gv6 | ICGV 03181 | Advanced line | ICRISAT-India |
| Gv7 | ICGV 03196 | Advanced line | ICRISAT-India |
| Gv8 | ICGV 03921 | Advanced line | ICRISAT-India |
| Gv9 | ICGV 07356 | Advanced line | ICRISAT-India |
| Gv10 | ICGV 07947 | Advanced line | ICRISAT-India |
| Gv11 | ICGV 13825 | Advanced line | ICRISAT-India |
| Gv12 | ICGV 13833 | Advanced line | ICRISAT-India |
| Gv13 | ICGV 13834 | Advanced line | ICRISAT-India |
| Gv14 | ICGV 14840 | Advanced line | ICRISAT-India |
| Gv15 | ICGV 14858 | Advanced line | ICRISAT-India |
| Gv16 | ICGV 14876 | Advanced line | ICRISAT-India |
| Gv17 | ICGV 55437 | Advanced line | ICRISAT-India |
| Gv18 | ICGV 86024 | Advanced line | ICRISAT-India |
| Gv19 | ICGV 86124 | Advanced line | ICRISAT-India |
| Gv20 | ICGV 89104 | Advanced line | ICRISAT-India |
| Gv21 | ICGV 91284 | Advanced line | ICRISAT-India |
| Gv22 | ICGV 91315 | Advanced line | ICRISAT-India |
| Gv23 | ICGV 92302 | Advanced line | ICRISAT-India |
| Gv24 | ICGV 93305 | Advanced line | ICRISAT-India |
| Gv25 | ICGV 94379 | Advanced line | ICRISAT-India |
| Gv26 | ICGV 94434 | Advanced line | ICRISAT-India |
| Gv27 | ICGV 96801 | Advanced line | ICRISAT-India |
| Gv28 | ICGV 96826 | Advanced line | ICRISAT-India |
| Gv29 | ICGV 96909 | Advanced line | ICRISAT-India |
| Gv30 | ICGV 97094 | Advanced line | ICRISAT-India |
| Gv31 | ICGV 97097 | Advanced line | ICRISAT-India |
| Gv32 | ICGV 97188 | Advanced line | ICRISAT-India |
| Gv33 | ICGV 98088 | Advanced line | ICRISAT-India |
| Gv34 | ICGV 98100 | Advanced line | ICRISAT-India |
| Gv35 | ICGV 99240 | Advanced line | ICRISAT-India |
| Gv36 | ICGV 99241 | Advanced line | ICRISAT-India |

Gv = Groundnut genotype.

## 2.4. Data collection

Data were collected both on plot and plant bases from the two rows for all traits following the standard description of groundnut [19]. Plant basis data collection (pods per plant and seeds per pod), five plants were randomly taken and the mean values of these five plants were calculated.

### 2.4.1. Phenological data.

**Days to 75% flowering (DF75) (days)**: It was recorded as the number of days from sowing to 75% of the plants in the plot started flowering.

**Days to 90% physiological maturity (DM90):** It was recorded as number of days from sowing to the stage when 90% of the plants in a plot have changed the color of their pods to dark.

**Grain filling period (GFP):** It was recorded as days from flowering to maturity, i.e., the number of days to maturity minus the number of days to 75% flowering.

### 2.4.2. Growth, yield and yield-related traits. *Growth traits.*

**Plant height (PH) (cm):** The mean height of main stem of five randomly taken plants was measured from the ground level or from the collar (point on the stem where roots start to grow).

**Number of branches per plant (NBP):** The mean number of branches per plant was obtained by counting the number of branches from each five sampled plants.

*Yield components traits.*

**Number of pods per plant (NPP):** This was determined as the mean value of five randomly sampled plants obtained by counting total number of pods per plant

**Number of seeds per pod (NSP):** The mean number of seeds per pod was obtained by counting the number of seeds collected from five mature pods from each five sampled plants.

**Hundred-kernel weight (HKW)**: Bulk of shelled seeds was counted and it was weighted and adjusted to standard moisture level (10%).

**Shelling percentage (SP) (%):** This was calculated by dividing the weight of seeds to the weight of total pods shelled expressed as percent.

$$\text{Shelling (\%)} = \frac{Seed\ weight}{Pod\ weight} \times 100$$

*Yield traits.*

**Pod yield (PY) (kg/ha):** The weight of dry pods from a unit plot was measured and was converted into kilogram per hectare.

**Kernel yield (KY) (kg ha$^{-1}$):** It was obtained from total harvest of the plot and adjusted to standard moisture level (10%) per plot in grams and converted into kilograms per hectare.

**Biomass yield (BY):** The weight in grams of sun dried above ground parts of the plants was recorded from the three rows and converted into kilogram per hectare.

**Harvest index (HI):** It was calculated as the ratio of kernel yield to total above ground biomass yield (biological yield).

$$\text{Harvest Index (HI)} = \frac{Kernel\ yield}{Total\ biomass} \times 100$$

*Derived yield trait.*

**Oil yield (OY):** It was calculated as oil content in percentage times kernel yield in ton/ ha.

### 2.5.  Data analysis

Prior to multivariate analysis, analysis of variance (ANOVA) was conducted to assess the presence of significant variation among genotypes.

**2.5.1.  Principal component analysis.**  Principal component analysis (PCA) was computed to find out the traits, which accounted more to the total variation. The data was standardized to mean zero and variance of one before computing principal component analysis to avoid differences in measurement scales. The principal component based on correlation matrix was calculated using SAS software version 9.0 [20]. Six principal components with eigenvalues greater than one were retained according to the Kaiser criterion, explaining 74.5% of the total variation.

**2.5.2.  Genetic divergence and clustering of genotypes.**  Genetic distance of 36 groundnut genotypes was estimated using Euclidean distance (ED) calculated from quantitative traits after standardization (subtracting the mean value and dividing it by the standard deviation) as established by [21] as follows:

$$ED_{jk} = \sqrt{\sum_{i=1}^{n}(X_{ij} - X_{ik})^2}$$

Where; $ED_{jk}$ = distance between genotypes j and k; $x_{ij}$ and $x_{ik}$ = phenotype traits values of the $i^{th}$ trait for genotypes j and k, respectively; and n = number of phenotype traits used to calculate the distance. The distance matrix from phenotype traits was used to construct dendrogram based on the Unweighted Pair-group Method with Arithmetic Means (UPGMA). The results of cluster analysis were presented in the form of dendrogram.

## 3.  Results

### 3.1.  Correlation among traits

The result of correlation matrix showed similar patterns of association at both genotypic and phenotypic levels, with genotypic correlations generally higher than phenotypic ones (Table 2). These higher genotypic associations indicate strong genetic control over trait associations rather than environmental control.

It was observed that, days to flowering showed very strong positive association with days to physiological maturity at both levels (rg = 0.97; rp = 0.93. This indicates uniform and coordinated reproductive progression. Grain filling period was positively associated with maturity period but negatively associated with shelling percentage, implying a potential trade-off between grain filling length and shelling efficiency. Among yield components, number of pods per plant showed strong positive association with number of branches per plant (rg = 0.40; rp = 0.40) and biological yield (rg = 0.58; rp = 0.53), indicating that branching contributes directly to high pod formation and biomass accumulation.

Dry pod yield showed strong positive association with kernel yield (rg = 0.90; rp = 0.87), confirming that kernel yield is primarily determined by pod productivity. Hundred kernel weight was also strongly associated with kernel yield (rg = 0.70; rp = 0.67), highlighting that the larger seed size directly contributes to higher yield. Oil yield showed almost perfect positive association with kernel yield (rg = 0.99; rp = 0.99), indicating that improvement in kernel yield directly enhances oil yield. Oil yield also exhibited strong association with dry pod yield, hundred kernel weight, biological yield, and branching traits. Branching increases reproductive sites, enhancing pod number and eventually oil yield. Genotypes producing more pods have the natural tendency to accumulate more oil per unit area. Strong association of oil yield with hundred kernel weight

**Table 2. Correlation matrix showing genotypic (above diagonal) and phenotypic (below diagonal) associations among measured traits in groundnut genotypes.**

| Traits | DTF | DTM | GFP | NPP | PH | NBP | BY | SP | DPY | KY | HKW | HI | OY |
|---|---|---|---|---|---|---|---|---|---|---|---|---|---|
| DTF | 1 | 0.97** | 0.35* | 0.01 | 0.17 | −0.08 | 0.06 | −0.14 | 0.04 | −0.03 | 0.14 | −0.24 | −0.06 |
| DTM | 0.93** | 1 | 0.45** | 0.03 | 0.19 | −0.01 | 0.03 | −0.19 | 0.07 | 0.00 | 0.13 | −0.28 | −0.02 |
| GFP | 0.31** | 0.41** | 1 | 0.07 | 0.00 | 0.20 | 0.00 | −0.36* | −0.01 | −0.11 | 0.04 | −0.18 | −0.09 |
| NPP | −0.08 | −0.11 | 0.05 | 1 | 0.20 | 0.40** | 0.58** | 0.16 | 0.31 | 0.38* | 0.39* | 0.32* | 0.40* |
| PH | 0.08 | 0.09 | 0.00 | 0.31** | 1 | 0.2 | 0.31 | 0.56** | 0.39* | 0.47** | 0.40* | 0.16 | 0.45** |
| NBP | −0.07 | −0.07 | 0.15 | 0.40** | 0.25* | 1 | 0.45** | 0.24 | 0.50** | 0.54** | 0.66** | 0.44** | 0.50** |
| BY | −0.02 | −0.05 | −0.02 | 0.53** | 0.27* | 0.44** | 1 | 0.37* | 0.50** | 0.60** | 0.57** | 0.32* | 0.61** |
| SP | −0.08 | −0.15 | −0.34** | 0.11 | 0.44** | 0.25* | 0.33** | 1 | 0.15 | 0.38* | 0.28 | 0.38* | 0.34* |
| DPY | 0.05 | 0.07 | −0.01 | 0.28* | 0.37** | 0.42** | 0.41** | 0.13 | 1 | 0.90** | 0.65** | 0.23 | 0.90** |
| KY | −0.03 | −0.01 | −0.11 | 0.31** | 0.42** | 0.48** | 0.53** | 0.35** | 0.87** | 1 | 0.70** | 0.37* | 0.99** |
| HKW | 0.13 | 0.13 | 0.04 | 0.29* | 0.34** | 0.56** | 0.46** | 0.22 | 0.60** | 0.67** | 1 | 0.45** | 0.72** |
| HI | −0.21 | −0.24* | −0.18 | 0.26* | 0.17 | 0.41** | 0.27* | 0.37** | 0.21 | 0.36** | 0.41** | 1 | 0.36* |
| OY | −0.06 | −0.03 | −0.09 | 0.30** | 0.39** | 0.46** | 0.52** | 0.32** | 0.86** | 0.99** | 0.68** | 0.34** | 1 |

indicates that the bigger, well-filled seeds contribute more oil per seed, increasing total oil yield. Genotypes with high bio-mass production have greater assimilate pool, supporting higher pod production and oil accumulation.

Negative associations among traits were generally weak, except for grain filling period with shelling percentage and days to physiological maturity with harvest index. Negative association between grain filling period and shelling percentage suggests that the extended grain filling period may favor shell growth more than kernel growth, reducing shelling percentage. Negative association occurred between days to physiological maturity and harvest index indicates that late-maturing genotypes invest more in vegetative structures, reducing partitioning efficiency to pods, thus lowering HI. Overall, the result of associations suggests that selection for high pod yield, larger seed size, and optimal branching will likely increase oil yield.

### 3.2. Principal component analysis

The principal component analysis (PCA) for 13 traits was computed to identify the critical traits which are important for the improvement of the crop and the traits that explained more of the variation in groundnut (Table 3). The results of PCA indicated that the first six principal components/factors were accounted for 74.50% of the total variance. Accordingly, the first principal component (PC1) or factor 1 (F1) accounted for approximately 29.77% of the total variation which was influenced positively by quantitative characters viz. kernel yield (KY), oil yield (OY), dry pod yield (DPY), hundred seed weight (HKW), biological yield (BY), number of branches per plant (NBP), plant height (PH), harvest index (HI), number of pods per plant (NPP) and shelling percentage (SP).

Principal component two (PC2) possessed an eigenvalue of 2.50 and showed 14.71% of total variation, which was largely influenced positively by only four quantitative characters viz. days to 75% flowering (DF75), days to physiological maturity (DM90), grain filling period (GFP) and hundred seed weight (HKW) whereas it was negatively influenced by HI in terms of factor loadings. This component discriminated the traits of genotypes based on differences in phenological development and yield-related attributes, particularly separating early and late maturing genotypes and highlighting variations in grain filling efficiency and kernel weight that influence overall productivity. Principal component three (PC3), with an eigenvalue of 1.73 explaining 10.16% of the total variation, was mainly characterized by positive contributions from SP, HI, and DPY, indicating their strong role in differentiating genotypes. This trait reflects a harvest partition efficiency, especially

**Table 3. Principal component values of the first six principal components from 13 traits of groundnut genotypes.**

| Trait | PC1 | PC2 | PC3 | PC4 | PC5 | PC6 |
|---|---|---|---|---|---|---|
| DTF | −0.07 | **0.90** | 0.07 | −0.05 | 0.13 | −0.03 |
| DTM | −0.08 | **0.94** | 0.003 | −0.05 | 0.13 | −0.001 |
| GFP | −0.10 | **0.56** | −0.24 | **0.40** | **−0.25** | **0.25** |
| NPP | **0.52** | −0.07 | 0.02 | **0.54** | 0.06 | **0.42** |
| PH | **0.55** | 0.17 | **0.40** | −0.08 | **0.36** | **0.32** |
| NBP | **0.66** | 0.04 | −0.04 | **0.41** | **−0.28** | −0.14 |
| BY | **0.68** | −0.005 | −0.07 | 0.20 | −0.12 | **0.26** |
| SP | **0.48** | −0.17 | **0.54** | **−0.23** | 0.06 | −0.02 |
| DPY | **0.79** | 0.16 | **−0.33** | −0.16 | 0.18 | −0.12 |
| KY | **0.91** | 0.05 | −0.22 | −0.21 | −0.10 | −0.13 |
| HSW | **0.76** | 0.24 | −0.09 | 0.06 | −0.10 | **−0.26** |
| HI | **0.54** | **−0.23** | **0.41** | 0.10 | **−0.37** | **−0.22** |
| OY | **0.89** | 0.03 | **−0.30** | **−0.24** | 0.08 | −0.10 |
| Eigenvalues | 5.04 | 2.50 | 1.73 | 1.33 | 1.04 | 1.00 |
| Proportions (%) | 29.77 | 14.71 | 10.16 | 7.82 | 6.14 | 5.90 |
| Cumulative (%) | 29.77 | 44.49 | 54.64 | 62.47 | 68.61 | 74.51 |

being driven by the factors SP, HI and DPY. It indicates that the trait discriminates genotypes in terms of the quantity partitioned towards economic yield and seed recovery, identifying those with better reproductive productivity.

In contrast, oil yield (OY) showed a negative loading, suggesting an inverse relationship with these traits and indicating that PC3 separates genotypes based on differences between seed productivity related traits and oil yield performance. This reflects a trade-off between pod productivity traits and oil yield, indicating differential assimilate allocation toward kernel mass versus oil accumulation among groundnut genotypes.

Principal component four (PC4) contained eigenvalues of 1.33 and contributed 7.82% of the total variation. The traits which contributed more positively to PC4 were NPP (0.54) followed by NBP (0.41) and (0.40) by GFP, while maximum negative value by OY (−0.24) followed by SP (−0.23). This component differentiated the genotypes based on phenology, productivity and quality, contrasting genoypes with high pod-setting and filling period against those with higher oil yield and shelling percentage. The fifth principal component (PC5) accounted for 6.14% of the variation with most of the variation being attributed to HI, PH and NBP. In PC6 highest positives were recorded for NPP (0.42), PH (0.32) and BY (0.26) on the other hand highest negative were recorded (−0.26) for HKSW, reflecting variation in vegetative and yield traits among genotypes.

### 3.3. Genetic divergence and clustering of genotypes

**3.3.1. Cluster analysis.** According to the result of cluster analysis, groundnut genotypes (n = 36) were grouped into six distinct clusters (Fig 1). Cluster-II comprised of 9 genotypes, which is about 25% of the total genotypes evaluated. Cluster III consisted eight (22.22%) of the total genotypes followed by cluster V which comprised seven (19.44%) genotypes. Cluster I consisted of six genotypes which is about 16.67% of the total genotypes tested. On the other hand, five genotype were included in clusters IV which accounted for 13.89% whereas cluster VI included only one genotype which is about 2.78% of total genotypes. The result implies that the crossing between distant genetic divergences of above diverse clusters might provide desirable recombinants for developing high yielding groundnut genotypes. This is because the cluster analysis demarcates genotypes into clusters, which exhibited high homogeneity within a cluster and high heterogeneity between clusters [22].

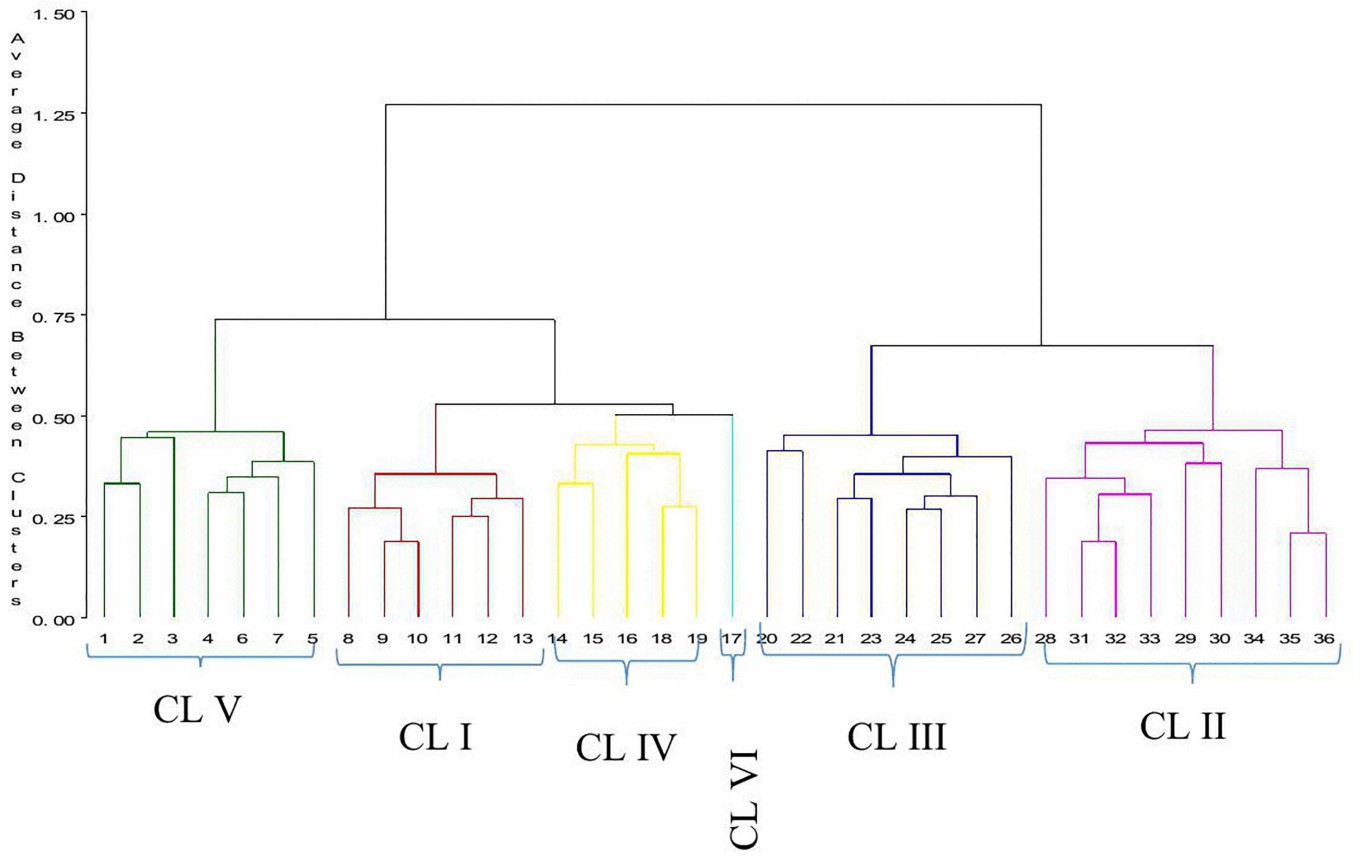

**Fig 1. Dendrogram illustrating genetic divergence of groundnut genotypes on the basis of 13 traits.**

**3.3.2. Cluster mean analysis.** There were noticeable variations between the six groups in the mean values of the 13 quantitative features (Table 4). With the exception of a few traits including GFP, NPP, NBP, and HKW, Cluster VI had lower mean values for most traits, whereas Cluster II typically had the highest mean values for the majority of traits, indicating greater yield and productivity. With certain features either above or below the overall mean, other clusters performed in an intermediate manner. All things considered, the clustering reveals significant genetic variation that can be used to select for and produce high-yielding groundnut lines.

**3.3.3. Genetic distance among groundnut genotypes.** The genetic distances for all possible pairs of 36 groundnut genotypes are presented in Appendix Table 1 in S1 Table. The genetic distances of genotypes varied from 2.45 to 8.54, with 5.44, 1.17, and 21.56% for the mean, standard deviation, and coefficient of variation, respectively (Table 5). The highest Euclidean distance between the groundnut genotypes was observed between Gv17 and Gv28 (8.54), followed by Gv3 and Gv23 (8.49), Gv3 and Gv30 (8.49), Gv15 and Gv17 (8.37), Gv22 and Gv28 (8.37) and Gv3 and Gv34 (8.36) (Appendix Table 1). While, the lowest genetic distance was found between Gv13 and Gv24 (2.45), and Gv14 and Gv36 (2.45), followed by Gv18 and Gv31 (2.65), Gv9 and Gv10 (2.83), Gv10 and Gv27(2.83), and Gv31 and Gv32 (2.83) (Appendix Table 1 in S1 Table).

The maximum Euclidean distance was obtained for genotype Gv17 (6.82) followed by Gv22 (6.46), Gv3 (6.44) and Gv30 (6.40), which also had Euclidean distance greater than the overall mean (5.44). Totally, 17 genotypes exhibited Euclidean distance greater than overall mean.

**Table 4. Mean values of six clusters for 13 traits of groundnut genotypes (n = 36).**

|  | CI | CII | CIII | CIV | CV | CVI | Overall mean |
|---|---|---|---|---|---|---|---|
| DTF | 55.33 | 55.22 | 54.88 | 57.40 | 57.64 | 55.0 | 55.93 |
| DTM | 114.58 | 115.39 | 114.94 | 117.80 | 118.21 | 116.0 | 116.06 |
| NPP | 36.18 | 37.77 | 39.80 | 31.62 | 33.41 | 50.80 | 36.62 |
| PH | 23.97 | 27.27 | 25.88 | 27.72 | 27.56 | 15.20 | 26.19 |
| NBP | 5.60 | 5.87 | 6.79 | 5.84 | 5.94 | 8.80 | 6.12 |
| BY | 4590.74 | 5620.53 | 5082.20 | 5420.00 | 4507.12 | 4469.4 | 5052.95 |
| SP | 65.67 | 69.22 | 64.63 | 69.30 | 66.43 | 61.00 | 66.85 |
| DPY | 3227.64 | 3622.89 | 3227.79 | 3659.62 | 3456.71 | 2254.8 | 3404 |
| KY | 2242.68 | 2616.96 | 2175.63 | 2631.85 | 2419.68 | 1446.7 | 2387.7 |
| HSW | 49.35 | 50.65 | 51.51 | 46.06 | 52.76 | 51.40 | 50.42 |
| HI | 47.23 | 45.82 | 43.17 | 49.77 | 53.52 | 30.83 | 47.09 |
| OY | 1.10 | 1.30 | 1.08 | 1.29 | 1.18 | 0.69 | 1.18 |

DTF = days to flowering, DTM = days to maturity, GFP = grain filling period, NPP = number of pods per plant, PH = plant height, NBP = number of branches per plant, BY= biomass yield, SP = shelling percentage, DPY = dry pod yield, KY = kernel yield, HSW = hundred seed weight, HI = harvest index, OY = oil yield.

**Table 5. Range and mean Euclidean distance of groundnut genotypes estimated from 13 quantitative traits (n = 36).**

| Gen. | Min | Max | Mean | SD | CV | Gen. | Min | Max | Mean | SD | CV |
|---|---|---|---|---|---|---|---|---|---|---|---|
| Gv1 | 3.46 | 6.93 | 5.22 | 0.93 | 17.82 | Gv20 | 3.87 | 7.28 | 5.60 | 0.86 | 15.30 |
| Gv2 | 4.12 | 7.81 | 6.16 | 1.02 | 16.50 | Gv21 | 3.32 | 7.07 | 4.74 | 0.99 | 20.85 |
| Gv3 | 4.00 | 8.49 | 6.44 | 1.15 | 17.88 | Gv22 | 4.24 | 8.37 | 6.46 | 1.00 | 15.50 |
| Gv4 | 3.46 | 7.28 | 5.24 | 1.06 | 20.15 | Gv23 | 3.46 | 8.49 | 5.57 | 1.09 | 19.48 |
| Gv5 | 3.74 | 8.06 | 5.92 | 0.91 | 15.35 | Gv24 | 2.45 | 6.48 | 4.64 | 0.86 | 18.64 |
| Gv6 | 3.87 | 8.06 | 5.69 | 1.07 | 18.72 | Gv25 | 3.61 | 6.86 | 5.26 | 1.00 | 19.09 |
| Gv7 | 3.46 | 7.42 | 5.22 | 1.04 | 19.85 | Gv26 | 4.36 | 7.35 | 5.77 | 0.71 | 12.38 |
| Gv8 | 3.46 | 7.35 | 5.13 | 1.07 | 20.88 | Gv27 | 2.83 | 6.93 | 4.79 | 1.15 | 24.09 |
| Gv9 | 2.83 | 6.56 | 4.63 | 1.02 | 21.97 | Gv28 | 3.32 | 8.54 | 5.58 | 1.29 | 23.20 |
| Gv10 | 2.83 | 7.48 | 5.11 | 1.15 | 22.62 | Gv29 | 4.24 | 8.06 | 6.11 | 0.83 | 13.59 |
| Gv11 | 3.32 | 6.78 | 4.80 | 0.92 | 19.21 | Gv30 | 4.69 | 8.49 | 6.40 | 0.82 | 12.88 |
| Gv12 | 3.00 | 7.07 | 4.92 | 0.81 | 16.36 | Gv31 | 2.65 | 7.35 | 4.63 | 1.13 | 24.29 |
| Gv13 | 2.45 | 6.48 | 4.85 | 0.84 | 17.42 | Gv32 | 2.83 | 7.87 | 5.31 | 1.22 | 22.96 |
| Gv14 | 2.45 | 7.81 | 5.25 | 1.02 | 19.48 | Gv33 | 3.16 | 7.62 | 5.45 | 1.14 | 20.92 |
| Gv15 | 4.36 | 8.37 | 6.22 | 0.82 | 13.13 | Gv34 | 3.74 | 8.37 | 5.92 | 1.05 | 17.66 |
| Gv16 | 3.16 | 8.00 | 5.37 | 1.34 | 24.97 | Gv35 | 3.00 | 6.86 | 4.68 | 0.78 | 16.57 |
| Gv17 | 4.36 | 8.54 | 6.82 | 0.94 | 13.81 | Gv36 | 2.45 | 6.86 | 4.93 | 0.95 | 19.18 |
| Gv18 | 2.65 | 7.07 | 5.06 | 0.93 | 18.40 | **Overall** | 2.45 | 8.54 | 5.44 | 1.17 | 21.56 |
| Gv19 | 3.74 | 8.00 | 5.89 | 1.14 | 19.31 |  |  |  |  |  |  |

Key. Gen. = Genotypes, Min = Minimum, Max = Maximum, SD = Standard Deviation, CV = Coeffici-ent of Variation.

## 4. Discussion

Traits interrelationship and multivariate analyses, viz. principal component analysis, UPGMA clustering and genetic distance analysis, were performed on a set of 36 groundnut genotypes by 13 traits, viz. days to flowering, days to physiological maturity, grain filling period, number of branches per plant, number of pods per plant, plant height, biological yield, shelling percentage, dry pod yield, hundred seed weight, harvest index, kernel yield and oil yield.

The correlation study suggested that the majority of features in groundnut were predominantly governed by genetic factors, as seen by the resemblance between genotypic and phenotypic correlations. Day to flowering was strongly correlated with day to maturity ($r_g$=0.97; $r_p$=0.93) indicating synchronized phenology, corresponding with recent Ethiopian research [23]. Yield components such as number of pods per plant, number of branches per plant, and biological yield were positively interrelated, confirming that enhanced vegetative growth supports reproductive capacity. These patterns align with findings from northwestern and western ethiopia, where pod number, kernel weight, and biomass were identified as major determinants of kernel yield [23,24,25].

Dry pod yield exhibited very strong association with kernel yield ($r_g$=0.90; $r_p$=0.87) and hundred kernel weight, while kernel yield was almost perfectly correlated with oil yield ($r_g$=0.99; $r_p$=0.99), indicating that improvement in these traits can simultaneously enhance kernel and oil productivity. The negative correlation between pod filling period and shelling percentage suggests potential trade-offs, also reported in local studies, that warrant careful selection. Overall, these findings support the use of pod number, dry pod yield, and kernel size as reliable indirect selection criteria for improving both kernel and oil yield in Ethiopian groundnut breeding programs.

Grouping genotypes based on their agro-morphological and quality characters is useful as it assists in identification and selection of best performers and genetically diverse parents for use in breeding program [5,26]. In the current study the principal component analysis revealed six components with eigenvalues greater than a unit. Principal components with eigenvalues greater than one are theoretically have more information than any single variable alone [27]. The first principal component (PC1), accounting for 29.77% of the total variance, was strongly influenced by yield and yield-related traits including kernel yield (KY), oil yield (OY), dry pod yield (DPY), hundred kernel weight (HKW), biological yield (BY), number of branches per plant (NBP), plant height (PH), harvest index (HI), number of pods per plant (NPP), and shelling percentage (SP). The clustering of these traits in PC1 reflects their coordinated contribution to overall productivity in groundnut genotypes. High positive loadings for these traits indicate that genotypes with higher biomass, more branches, larger seeds, and higher pod numbers tend to produce higher kernel and oil yield. This demonstrates that PC1 essentially represents a "yield and productivity component", capturing the major sources of variation directly linked to economic output, which is typical in groundnut where seed yield is determined by the combined effect of vegetative growth and reproductive traits. This pattern is consistent with recent groundnut research in Ethiopia, where multivariate analysis identified similar yield components as the main contributors to variation among genotypes evaluated under diverse agro-ecologies [9]. In contrast, PC2 explained 14.71% of the total variance and was primarily influenced by phenological traits such as days to 75% flowering (DF75), days to physiological maturity (DM90), and grain filling period (GFP) together with hundred kernel weight (HKW), while harvest index (HI) had a negative loading. This pattern indicates that PC2 reflects variation in crop development and maturity, distinguishing early versus late maturing genotypes. Positive loadings for flowering and maturity traits suggest that genotypes with longer growth durations tend to have longer grain-filling periods, potentially enhancing kernel size (HKW). The negative influence of HI implies that prolonged phenology may not necessarily translate to higher efficiency in converting biomass to kernel yield. Therefore, PC2 captures the temporal and developmental dimension of variation among genotypes, which is important for adaptation and selection in specific environments. Positive loadings for flowering and maturity traits align with findings from Ethiopian genotype evaluations demonstrating that phenological differences significantly differentiate groundnut lines adapted to varying rainfall and temperature regimes in Northern and Eastern Ethiopia [28,29]. The cluster analysis grouped the 36 groundnut genotypes into six distinct clusters, indicating the presence of considerable genetic diversity

among the evaluated material. The result suggests that some genotypes share close genetic resemblance, whereas others are highly divergent. This pattern reflects a structured germplasm, where genotypes within the same cluster are likely to respond similarly to local environmental conditions, implying that selection of representatives from each cluster could capture both stability and broad adaptability, and the presence of highly divergent genotypes indicates potential for introducing novel traits, such as stress tolerance or superior yield components, into breeding populations. Crossing genotypes from clusters that are genetically distant (e.g., Cluster VI with Cluster II or III) could maximize heterosis and generate segregating populations with broad variability. This pattern is consistent with recent Ethiopian groundnut characterization studies, which reported that landraces and breeding lines from contrasting agro-ecologies often cluster separately due to distinctive phenological and yield trait profiles [30,28]. In those studies, genotypes adapted to drought-prone lowlands differed markedly from those selected under higher moisture conditions, suggesting that environmental adaptation history contributes significantly to genetic divergence in Ethiopian groundnut germplasm. Groundnut genotypes grouped in different clusters could be evaluated for combining ability to constitute a pool of best parents. These findings are supported by previous reports by [31]; and [32] that there is high genetic diversity in groundnut. The grouping of the genotypes indicated that evaluated characters had influence on clustering pattern. Similar result was reported in groundnut by [33]; and [34].

The genetic distances among the 36 groundnut genotypes exhibited substantial variability, ranging from 2.45 to 8.54, with a mean of 5.44, a standard deviation of 1.17, and a coefficient of variation of 21.56%. Nearly half of the genotypes (47.2%) displayed mean genetic distances exceeding the overall average, indicating considerable genetic divergence within the studied materials. This wide genetic variability is consistent with earlier findings by [9] and [26], who reported significant clustering and genetic distance variation among Bambara groundnut germplasms. These results suggested that the genotypes that had greater mean of genetic distance over the average mean distance were more distant to others and crossing between these genotypes could be used to combine desired traits in progeny of subsequent generation. Thus, the maximum amount of heterosis, which could have implication for genetic improvement of the crop for the target trait, is expected from the crosses with genotypes that are distant from each other. Based on the result of the genetic distance analysis using Euclidean method, there were variations of genetic distances among genotypes, Gv17 and Gv28, Gv3 and Gv23, Gv3 and Gv30, Gv15 and Gv17, Gv22 and Gv28, and Gv3 and Gv34. Crossing these genetically distant parents is expected to enhance variability, facilitate the combination of desirable traits, and accelerate genetic improvement for target traits in groundnut. Therefore, the future research should focus on:

- Conducting diallel or line × tester crossing experiments to validate heterosis and combining ability among the identified divergent genotypes.

- Evaluating the resulting progenies across multiple environments to assess stability and genotype × environment interactions.

- Incorporating molecular marker analysis to complement phenotypic distance estimates and confirm genetic divergence at the DNA level.

- Investigating the inheritance patterns of key economic traits to enhance selection efficiency in subsequent breeding cycles.

## Supporting information

**S1 Table. Euclidean distances of groundnut genotypes estimated from mean values of genotypes for 13 quantitative traits (n = 36).**
(DOCX)

**S1 File. Raw dataset of agro-morphological and quality traits of groundnut (*Arachis hypogaea* L.) genotypes evaluated in Eastern Ethiopia.** This dataset was used for multivariate statistical analysis including PCA, cluster analysis, and genetic distance analysis.
(XLSX)

## Acknowledgments

The authors wish to express their profound appreciation to the Dire Dawa, Tony farm Research sub-Station of Haramaya University for generously providing the field experimental facilities and technical support necessary for the successful completion of this study. The authors are also deeply gratitude to the reviewers for their insightful comments and constructive suggestions, which substantially enhanced the scientific quality and clarity for the manuscript.

## Author contributions

**Conceptualization:** Abdi Mohammed Hassen.

**Writing – original draft:** Desu Beriso Dama.

**Writing – review & editing:** Seltene Abadi Tesfamariam.

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
