## [Decision Letter · Decision Letter 0]

3 Feb 2026

PONE-D-26-00818Multivariate Analysis for Agro-Morphological and Quality Traits in Groundnut (Arachis hypogaea L.) Genotypes in Eastern EthiopiaPLOS One

Dear Dr. Dama,

Thank you for submitting your manuscript to PLOS ONE. After careful consideration, we feel that it has merit but does not fully meet PLOS ONE’s publication criteria as it currently stands. Therefore, we invite you to submit a revised version of the manuscript that addresses the points raised during the review process.

If applicable, we recommend that you deposit your laboratory protocols in protocols.io to enhance the reproducibility of your results. Protocols.io assigns your protocol its own identifier (DOI) so that it can be cited independently in the future. For instructions see: https://journals.plos.org/plosone/s/submission-guidelines#loc-laboratory-protocols. Additionally, PLOS ONE offers an option for publishing peer-reviewed Lab Protocol articles, which describe protocols hosted on protocols.io. Read more information on sharing protocols at . Additionally, PLOS ONE offers an option for publishing peer-reviewed Lab Protocol articles, which describe protocols hosted on protocols.io. Read more information on sharing protocols at https://plos.org/protocols?utm_medium=editorial-email&utm_source=authorletters&utm_campaign=protocols..

We look forward to receiving your revised manuscript.

Kind regards,

Sumit Jangra, Ph.D.

Academic Editor

PLOS One

Journal Requirements:

3. We note that your Data Availability Statement is currently as follows: “All relevant data are within the manuscript and its Supporting Information files”

Reviewers' comments:

Reviewer's Responses to Questions

**Comments to the Author**

1. Is the manuscript technically sound, and do the data support the conclusions?

Reviewer #1: Yes

Reviewer #2: Yes

2. Has the statistical analysis been performed appropriately and rigorously? 

Reviewer #1: Yes

Reviewer #2: Yes

3. Have the authors made all data underlying the findings in their manuscript fully available?

Reviewer #1: Yes

Reviewer #2: Yes

4. Is the manuscript presented in an intelligible fashion and written in standard English?

Reviewer #1: Yes

Reviewer #2: No

5. Review Comments to the Author

Reviewer #1: The MS Multivariate Analysis for Agro-Morphological and Quality Traits in Groundnut (Arachis hypogaea L.) Genotypes in Eastern Ethiopia needs extensive correction and improvement. Pl see the attachment.

Reviewer #2: • The manuscript addresses an important topic in crop improvement, namely the assessment of genetic variability in groundnut using multivariate approaches. The use of principal component analysis, clustering, and genetic distance to characterize diversity among genotypes is relevant and appropriate for breeding-oriented research. The experimental design (simple lattice) and the inclusion of multiple agronomic and quality traits are also strengths of the manuscript.

• However, in its current form, the manuscript suffers from major weaknesses related to scientific presentation, clarity of writing, and overall coherence. Substantial revision is required before it can be considered for publication.

• The manuscript requires thorough editing by a fluent English speaker or a professional language editing service before resubmission, as it contains pervasive grammatical errors, unclear sentences, and inappropriate wording that significantly affect readability and professionalism. Examples include incorrect phrases such as “Those 36 genotypes were grouped…”, “writing curative data analysis…”, “We are highly acknowledged…”, and “Declaration of Computing Interest.” These issues occur throughout the manuscript (Abstract, Discussion, Author Contributions, Acknowledgments, etc.).

• The Abstract is overly descriptive and contains grammatical problems, redundant wording, and weak articulation of novelty and implications. The opening sentence reads more like a textbook-style introduction than a research-driven statement. Although the findings report statistical outputs (e.g., PCA variance, clusters), they do not explain why these results matter biologically. The abstract should be rewritten to include: a clear background (1–2 sentences), specific objective, brief methodology, concise key results, and practical implications for breeding.

• The Introduction provides relevant background but is overly descriptive, poorly structured, and lacks a clearly articulated research gap. Excessive textbook-level information should be condensed, and the narrative reorganized to progress logically from general context to specific problem. For example, there is poor logical flow between paragraphs: the Introduction jumps between global production, nutrition, constraints, genetics, and diversity without a smooth conceptual progression. A stronger structure would be: brief importance of groundnut, challenges in Ethiopia, importance of genetic variability for improvement, knowledge gap in Ethiopian germplasm, and purpose of the current study.

• The final paragraph (e.g., “Accordingly, the study was used multivariate analysis to assess…”) attempts to present the aim, but the language is grammatically weak and scientifically imprecise, which weakens the academic tone. The objective should be rewritten clearly and concisely, ideally in one or two strong sentences. For example:

“Therefore, the objective of this study was to assess genetic variability among 36 groundnut genotypes using agro-morphological and quality traits through multivariate analyses (PCA, clustering, and genetic distance) in eastern Ethiopia.”

In addition, citation accuracy and relevance must be carefully reviewed.

• Several equations are poorly formatted and should be standardized.

• The Results and Discussion sections are overly descriptive and weakly analytical. In the Discussion, there is limited critical interpretation of why certain traits contributed strongly to PC1 and PC2, what the observed clustering implies for adaptation, breeding strategy, or germplasm structure, and why particular genotypes (e.g., Gv3, Gv17, Gv28) are highly divergent.

• The Discussion should be strengthened by interpreting the PCA results biologically rather than only statistically, linking the findings to breeding objectives in Ethiopia, and explaining how the results advance existing knowledge rather than merely confirming previous studies.

• The Author Contributions statement should be rewritten for clarity and professionalism.

• The Acknowledgments section requires grammatical correction.

• Keywords should avoid redundancy with the title and be more specific.

• The manuscript contains several inconsistencies between in-text citations and the reference list. Key cited works (e.g., Amare et al., 2017; Niveditha et al., 2016; Chahal & Gosal, 2002; Yan & Tinker, 2005) are missing from the reference list. Conversely, some references listed are not cited in the text. In addition, discrepancies exist in publication years (e.g., FAOSTAT cited as 2022 in-text but listed as 2020 in the references; SAS cited as 2000 in-text but listed as 2004 in the reference list). These issues must be carefully corrected to meet journal standards.

6. PLOS authors have the option to publish the peer review history of their article (what does this mean?). If published, this will include your full peer review and any attached files.). If published, this will include your full peer review and any attached files.

.

Reviewer #1: No

Reviewer #2: No

---

## [Author Response · Author response to Decision Letter 1]

20 Mar 2026

All requested issues have been addressed. The author information has been updated in the submission system to include Desu Beriso, Seltene Abadi, and Abdi Mohammed. References to Table 3 and Table 5 have been added in the manuscript text. The abstract in the manuscript and the abstract in the submission system have been made identical. In addition, the minimal dataset underlying the results has been uploaded as a Supporting Information file and a caption has been added at the end of the manuscript.

---

## [Decision Letter · Decision Letter 1]

8 Apr 2026

Multivariate Analysis for Agro-Morphological and Quality Traits in Groundnut (Arachis hypogaea L.) Genotypes in Eastern Ethiopia

PONE-D-26-00818R1

Dear Dr. Dama,

We’re pleased to inform you that your manuscript has been judged scientifically suitable for publication and will be formally accepted for publication once it meets all outstanding technical requirements.

An invoice will be generated when your article is formally accepted. Please note, if your institution has a publishing partnership with PLOS and your article meets the relevant criteria, all or part of your publication costs will be covered. Please make sure your user information is up-to-date by logging into Editorial Manager at Editorial Manager® and clicking the ‘Update My Information' link at the top of the page. For questions related to billing, please contact  and clicking the ‘Update My Information' link at the top of the page. For questions related to billing, please contact billing support..

Kind regards,

Sumit Jangra, Ph.D.

Academic Editor

PLOS One

Additional Editor Comments (optional):

Reviewers' comments:

Reviewer's Responses to Questions

**Comments to the Author**

1. If the authors have adequately addressed your comments raised in a previous round of review and you feel that this manuscript is now acceptable for publication, you may indicate that here to bypass the “Comments to the Author” section, enter your conflict of interest statement in the “Confidential to Editor” section, and submit your "Accept" recommendation.

Reviewer #1: (No Response)

Reviewer #2: All comments have been addressed

2. Is the manuscript technically sound, and do the data support the conclusions?

Reviewer #1: Yes

Reviewer #2: Yes

3. Has the statistical analysis been performed appropriately and rigorously? 

Reviewer #1: Yes

Reviewer #2: Yes

4. Have the authors made all data underlying the findings in their manuscript fully available?

Reviewer #1: Yes

Reviewer #2: Yes

5. Is the manuscript presented in an intelligible fashion and written in standard English?

Reviewer #1: Yes

Reviewer #2: (No Response)

6. Review Comments to the Author

Reviewer #1: The authors have incorporated all the suggestions; therefore, the manuscript may be considered for publication. However, some latest findings in discussion section still missing, During final proofreading, the authors must add below mentioned lateset findinding in discussion to strength the discussion part particularly correlation and PCA and also follow the journal’s formatting guidelines and ensure that botanical names are written in italics.

1. https://doi.org/10.1080/23311932.2025.2610022

2. https://doi.org/10.1186/s12870-025-06335-x

3. https://doi.org/10.1016/j.indcrop.2026.123162

4. https://doi.org/10.1186/s12870-025-06985-x

5. https://doi.org/10.1038/s41598-023-49091-4

Reviewer #2: (No Response)

7. PLOS authors have the option to publish the peer review history of their article (what does this mean?). If published, this will include your full peer review and any attached files.). If published, this will include your full peer review and any attached files.

.

Reviewer #1: **Yes:**Dr Lalu Prasad Yadav, Senior Scientist, ICAR-CIAH, Bikaner, IndiaDr Lalu Prasad Yadav, Senior Scientist, ICAR-CIAH, Bikaner, India

Reviewer #2: No

---

## [Editor Report · Acceptance letter]

PONE-D-26-00818R1

PLOS One

Dear Dr. Dama,

I'm pleased to inform you that your manuscript has been deemed suitable for publication in PLOS One. Congratulations! Your manuscript is now being handed over to our production team.

Kind regards,

on behalf of

Dr. Sumit Jangra

Academic Editor

PLOS One